# Non-Parametric Statistic for Testing Cumulative Abnormal Stock Returns

**Seppo Pynnonen**

Department of Mathematics and Statistics, University of Vaasa, P.O. Box 700, FI-65101 Vaasa, Finland; sjp@uwasa.fi; Tel.:+358-21-449-8311

**Abstract:** Due to the non-normality of stock returns, nonparametric rank tests are gaining accceptance relative to parametric tests in financial economics event studies. In rank tests, financial assets' multiple day cumulative abnormal returns (CARs) are replaced by cumulated ranks. This paper proposes modifications to the existing approaches to improve robustness to cross-sectional correlation of returns arising from calendar time overlapping event windows. Simulations show that the proposed rank test is well specified in testing CARs and is robust towards both complete and partial overlapping event windows.

**Keywords:** finance; economics; event study; clustered event days; cross-sectional correlation; cumulated ranks; rank test; standardized abnormal returns

**JEL Classification:** G14; C10; C15

## 1. Introduction

Efficient markets has been and still is a cornerstone of asset pricing theory. Empirical work in this regard is largely concerned with the adjustment of security prices to relevant information. Fama (1970, 1991) refine relevant information into three hierarchical subsets of weak form, semi-strong form, and strong form Fama (1970), or equivalently, return predictability, event studies, and private information Fama (1991). Event studies investigate the effect of unexpected economic events on asset prices. Therefore, event studies can give the most direct evidence on market efficiency (c.f. Fama 1991, p. 1577). For this purpose, asset price data available from financial markets can be used with appropriate statistical testing methodology, reliability of which is central in inferences. In order to foster this, the current paper aims to fill the gap in existing (non-parametric) statistical testing by proposing non-parametric rank tests that are robust to cross-sectional dependency of asset returns in more general circumstances than the existing ones. Otherwise, refer to (Campbell et al. 1997, chp. 4) as an excellent introduction to event studies and related statistical methods.

Regarding methodology, standardizing returns by their respective standard deviations homogenizes data and has proven to improve testing performance. Because of this improvement, standardized return based tests by Patell (1976) and Boehmer et al. (1991) (BMP) have gained popularity over conventional non-standardized tests in testing event effects on mean security price performance. Harrington and Shrider (2007) found that in a short-horizon testing of abnormal returns (i.e., systematic deviation from expected behavior), one should always use methods that are robust to cross-sectional variation in the true abnormal returns.[1] They found that BMP is a good candidate for robust, parametric tests in conventional event studies.[2]

A major problem in statistical tests of returns is that the returns are not normally distributed (Fama 1976). Not surprisingly, non-parametric rank tests introduced by Corrado (1989, 2011); Corrado and Zivney (1992); Campbell and Wasley (1993) and Kolari and

Pynnonen (2011), among others, dominate parametric tests both in terms of better size and power (e.g., see Campbell and Wasley 1993; Corrado 1989; Corrado and Zivney 1992; Kolari and Pynnonen 2011; Kolari and Pynnönen 2010; Luoma 2011). Furthermore, rank tests by Corrado and Zivney (1992) and Kolari and Pynnonen (2011) that utilize event period re-standardized returns have proven to be robust to event-induced volatility (Kolari and Pynnonen (2011); Kolari and Pynnönen (2010)), cross-correlation due to event day clusterings (Kolari and Pynnönen 2010), and autocorrelation (Kolari and Pynnonen 2011). These studies are consistent with the view stated in the epilogue of Lehmann (2006): "Rank tests apply often to relatively simple solutions, such as one-, two-, and *s*-sample problems, and testing for independence and randomness, but for these situations they are often the method of choice". (Lehmann 2006, p. v). In addition, the results of rank tests are invariant to monotone transformations of the underlying returns; that is, whether the returns are defined as simple, continuously compounded log returns.

Existing rank based tests, however, are not robust to cross-sectional correlation if the event days are partially overlapping in calendar time. This *partial clustering* occurs when events are in calendar time scattered within an event window more or less randomly rather than clustered on the same calendar day (i.e., *complete clustering* as in Kolari and Pynnönen 2010). Figure 1 illustrates the various degrees of clustering in terms of three stocks. Panel A depict the non-clustered case, Panel B the partial clustering, and Panel C the complete clustering. In the complete clustering the event days are the same in calendar time, while in the partial clustering the event days may or may not be the same in calendar time but the event windows are more or less overlapping. In the non-clustered case the event windows are completely separate in calendar time. In this case all event effects can be investigated utilizing cross-sectional independence assumption of returns. In complete clustering cross-sectional correlation of returns must be fully accounted for. In the partial case the correlation can bias the results depending on the degree of overlapping. For example in the case of Panel C if the interest is only on the event day effect, as all the event days are different, there is no biasing effect by the correlation. On the other hand, if cumulative return effect over the whole event window is of interest, correlation of returns on the overlapping affects the joint behavior of the cumulative returns.

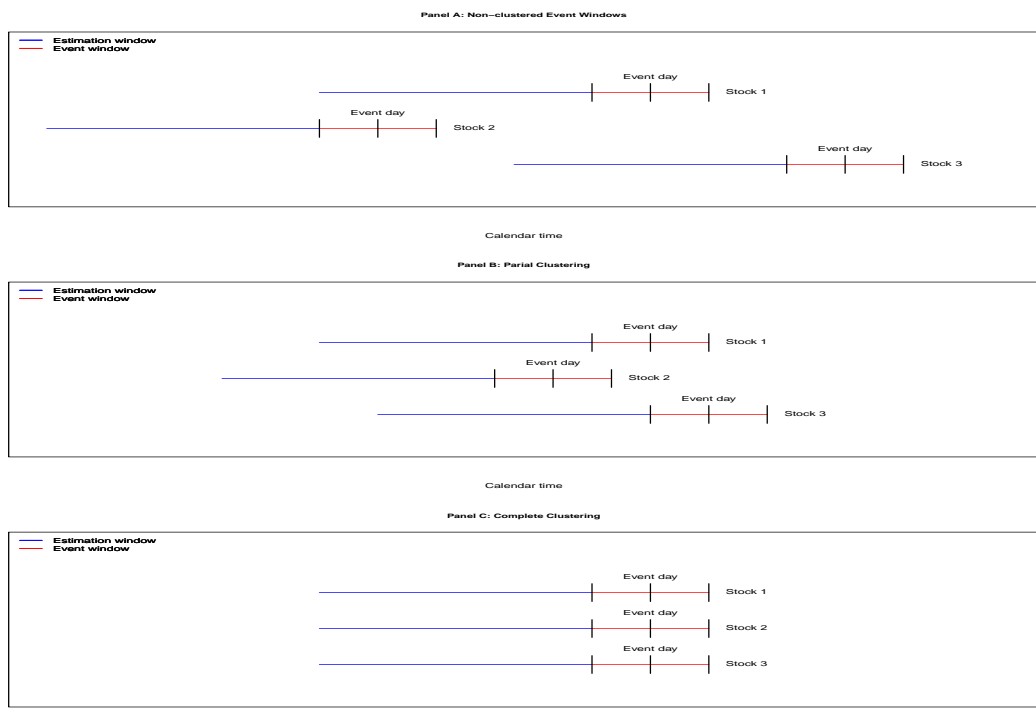

**Figure 1.** Event Windows Clustering.



Jaffe (1974) is probably the first paper in event study testing to address the potential biasing effect of cross-sectional correlation due to clustered events. Table 2 of Kolari and Pynnönen (2010) explicitly addresses the issue by showing that already a virtually trivial cross-sectional correlation, such as 0.05, can severely bias testing for event effects towards material over-rejection. The present paper seeks to fill this gap of accounting for cross-sectional correlation in non-parametric even study testing also with partially clustered event days.

The paper is organized as follows. Section 2 reviews some related key literature. Section 3 defines the main concepts and derives some distributional properties of rank statistics. Section 4 introduces the new transformed rank test. Section 5 reports simulation results, and Section 6 concludes.

## 2. Review of Related Literature

Patell and BMP parametric tests are straightforward tests of cumulative abnormal returns (CARs) over multiple day windows. With the correction suggested by Kolari and Pynnönen (2010), these tests are useful in the case of completely clustered event days, and with the correction suggested by Kolari et al. (2018) when event days are either completely or partially clustered. By construction, the Corrado (1989) non-parametric rank test applies for testing single day event returns. Testing for CARs with the same logic implies the need of defining multiple-day returns that match the number of days in the CARs, (see (Corrado 1989, p. 395); (Campbell and Wasley 1993, footnote 4)). In practice this approach is carried out by dividing the estimation period and event period into intervals matching the number of days in the CAR. Unfortunately, this procedure is not useful for a number of reasons. Foremost among these is that it does not necessarily lead to a unique testing procedure. In addition, the abnormal return model should be re-estimated for each multiple-day CAR definition. Furthermore, for a fixed estimation period, as the number of days accumulated in a CAR increases, the number of multiple-day estimation period observations reduces quickly impractically low and thus would weaken the abnormal return model estimation (c.f., Kolari and Pynnönen 2010). Kolari and Pynnonen (2011) solve these issues in their generalized rank test approach.

On the other hand, Campbell and Wasley (1993) recommend using the Corrado (1989) rank test to test cumulative abnormal returns by simply accumulating the respective ranks to constitute cumulative ranks (see also Hagnäs and Pynnonen 2014). This practice is adopted in the Eventus® software Cowan (2007) and is probably, for the time being, the most popular procedure for multiple day applications of rank tests. An advantage is that this proceure implicitly accounts the cross-sectional correlation in the case of the complete clustering.

In spite of these attractive properties, the cumulative ranks test does not account for cross-sectional correlation due to calendar time partially overlapping event windows, i.e., the case of partial clustering. As referred above, even a small (positive) correlation biases the standard errors downwards leading to over-rejection of the null hypothesis of no event effect. Contributing to the event study literature, this paper proposes an adjustment for the standard errors that corrects the bias in non-parametric testing.

## 3. Distributional Properties of Ranks

We begin by fixing some notations and an underlying assumption to facilitate our theoretical discussion.

**Assumption 1.** *Stock returns $r_{it}$ for firm i are weak white noise continuous random variables and are cross-sectionally independent over non-overlapping calendar days, or,*

$$
\begin{aligned}
\mathbb{E}[r_{it}] &= \mu_i \text{ for all } t \\
\mathrm{var}[r_{it}] &= \sigma_i^2 \text{ for all } t \\
\mathrm{cov}[r_{it}, r_{iu}] &= 0 \text{ for all } t \neq u \\
r_{it} \text{ and } \quad r_{ju} &\quad \text{are independent whenever } i \neq j \text{ and } t \neq u.
\end{aligned}
\tag{1}
$$

It is a stylized fact that the variances of the returns are time varying and that there is mild autocorrelation. The time varying volatility problem can be partially captured in terms of GARCH-modeling. However, typical GARCH-processes satisfy Assumption 1.

Let $\mathrm{AR}_{it} = r_{it} - \mathbb{E}[r_{it}]$ denote the abnormal return of security $i$ on day $t$, and following commonly used notations (e.g., Brown and Warner 1985, p. 6), let day $t = 0$ indicate the event day. Days from $t = T_0 + 1$ to $t = T_1$ represent the estimation period relative to the event day, and days from $t = T_1 + 1$ to $t = T_2$ represent the event window. The cumulative abnormal return (CAR) from $\tau_1$ to $\tau_2$ with $T_1 < \tau_1 \leq \tau_2 \leq T_2$, is defined as

$$\mathrm{CAR}_i(\tau_1, \tau_2) = \sum_{t=\tau_1}^{\tau_2} \mathrm{AR}_{it}. \tag{2}$$

The time period from $\tau_1$ to $\tau_2$ is called in the following as a CAR window or CAR period.

Standardized abnormal returns are defined as

$$\mathrm{SAR}_{it} = \frac{\mathrm{AR}_{it}}{S_{\mathrm{AR}_i}}, \tag{3}$$

where

$$S_{\mathrm{AR}_i} = \sqrt{\frac{1}{T_1 - T_0 - 1} \sum_{t=T_0+1}^{T_1} \mathrm{AR}_{it}^2}. \tag{4}$$

Furthermore, for the purpose of accounting the possible event induced volatility, the re-standardized abnormal returns are defined as in Boehmer et al. (1991) (see also, Corrado and Zivney 1992), or

$$\mathrm{SAR}'_{it} = \begin{cases} \mathrm{SAR}_{it} / S_{\mathrm{SAR}_t}, & T_1 < t \leq T_2 \\ \mathrm{SAR}_{it}, & \text{otherwise,} \end{cases} \tag{5}$$

where

$$S_{\mathrm{SAR}_t} = \sqrt{\frac{1}{n-1} \sum_{i=1}^{n} (\mathrm{SAR}_{it} - \overline{\mathrm{SAR}}_t)^2} \tag{6}$$

is the time $t$ cross-sectional standard deviation of $\mathrm{SAR}_{it}$, $\overline{\mathrm{SAR}}_t = \frac{1}{n} \sum_{i=1}^{n} \mathrm{SAR}_{it}$, and $n$ is the number of stocks in the portfolio. In addition, let $K_{it}$ denote the rank numbers of abnormal returns, where $K_{it} \in \{1, \ldots, T\}$, $t = T_0 + 1, \ldots, T_2$, $T = T_2 - T_0$, and $i = 1, \ldots, n$.

If the available observations in the estimation period vary from one series to another, it is convenient to use *standardized ranks* with zero mean and unit variance. To do this, we compile the known results of rank statistics (e.g., Lehmann 2006, Appendix, Section 1) as described below.

**Result 1.** *Let $K_{it}$ denote the rank numbers as defined above, then*

$$\begin{align} \mathbb{E}[K_{it}] &= (T+1)/2 \tag{7} \\ \mathrm{var}[K_{it}] &= (T^2 - 1)/12 \tag{8} \\ \mathrm{cov}[K_{is}, K_{it}] &= -(T+1)/12, (s \neq t). \tag{9} \end{align}$$

**Definition 1.** *Standardized ranks are defined as*

$$U_{it} = \frac{K_{it} - \frac{1}{2}(T+1)}{\sqrt{(T^2 - 1)/12}}. \tag{10}$$

(c.f., Hagnäs and Pynnonen 2014).

By Result 1, we obtain:

**Result 2.**

$$
\begin{align}
\mathbb{E}[U_{it}] &= 0 \tag{11} \\
\text{var}[U_{it}] &= 1 \tag{12} \\
\text{cov}[U_{is}, U_{it}] &= -1/(T-1). \tag{13}
\end{align}
$$

Next, we define *cumulative standardized ranks* for individual stocks.

**Definition 2.** *The cumulative standardized ranks of a stock i over the event days window form $\tau_1$ to $\tau_2$ are defined as*

$$
U_i(\tau_1, \tau_2) = \sum_{t=\tau_1}^{\tau_2} U_{it}, \tag{14}
$$

*where $T_1 < \tau_1 \leq \tau_2 \leq T_2$.*

From Result 2 and utilizing the variance-of-the-sum formula, $\text{var}[U_i(\tau_1, \tau_2)] = \sum_{t=\tau_1}^{\tau_2} \text{var}[U_{it}] + \sum_{s \neq t} \text{cov}[U_{is}, U_{it}]$, we obtain:

**Result 3.**

$$
\begin{align}
\mathbb{E}[U_i(\tau_1, \tau_2)] &= 0 \tag{15} \\
\text{var}[U_i(\tau_1, \tau_2)] &= \frac{\tau(T-\tau)}{T-1}, \tag{16}
\end{align}
$$

*where $i = 1, \ldots, n$, $T_1 < \tau_1 \leq \tau_2 \leq T_2$, and $\tau = \tau_2 - \tau_1 + 1$.*

Rather than investigating individual (cumulative) returns, the practice in event studies is to aggregate individual returns into equally-weighted portfolios such that:

**Definition 3.** *The average cumulative standardized ranks are defined as the equally weighted portfolio of individual cumulative standardized ranks defined in* (14), *i.e.,*

$$
\bar{U}(\tau_1, \tau_2) = \frac{1}{n} \sum_{i=1}^{n} U_i(\tau_1, \tau_2) = \sum_{t=\tau_1}^{\tau_2} \bar{U}_t, \tag{17}
$$

*where $T_1 < \tau_1 \leq \tau_2 \leq T_2$ and*

$$
\bar{U}_t = \frac{1}{n} \sum_{i=1}^{n} U_{it} \tag{18}
$$

*is the time t average of standardized ranks.*

The expected value of $\bar{U}(\tau_1, \tau_2)$ is the same as that of the cumulative ranks of individual securities, or

$$
\mathbb{E}[\bar{U}(\tau_1, \tau_2)] = \frac{1}{n} \sum_{i=1}^{n} \mathbb{E}[U_i(\tau_1, \tau_2)] = 0.
$$

If the event days are not clustered the cross-sectional correlations of the return series are zero (or at least negligible). Under the cross-sectional independence and by Equation (16), the variance of $\bar{U}(\tau_1, \tau_2)$ is

$$
\sigma_\tau^2 = \text{var}[\bar{U}(\tau_1, \tau_2)] = \frac{\tau(T-\tau)}{(T-1)n}. \tag{19}
$$

Then by the central limit theorem

$$Z = \left( \frac{(T-1)n}{\tau(T-\tau)} \right)^{\frac{1}{2}} \bar{U}(\tau_1, \tau_2) \sim N(0,1) \text{ as } n \to \infty. \tag{20}$$

The situation is more complicated if the event days are partially overlapping in calendar time which implies cross-sectional correlation. Recalling that the variances of $U_i(\tau_1, \tau_2)$ given in Equation (16) are constants (independent of $i$), we can write the cross-sectional covariance of $U_i(\tau_1, \tau_2)$ and $U_j(\tau_1, \tau_2)$ as

$$\text{cov}\left[ U_i(\tau_1, \tau_2), U_j(\tau_1, \tau_2) \right] = \frac{\tau(T-\tau)}{T-1} \rho_{ij}(\tau_1, \tau_2), \tag{21}$$

where $\rho_{ij}(\tau_1, \tau_2)$ is the cross-sectional correlation of $U_i(\tau_1, \tau_2)$, and $U_j(\tau_1, \tau_2)$, $i, j = 1, \ldots, n$. Utilizing this result and the variance-of-the-sum formula, the variance of $\bar{U}(\tau_1, \tau_2)$ in (17) becomes:

**Result 4.**

$$\begin{aligned}
\text{var}[\bar{U}(\tau_1, \tau_2)] &= \frac{1}{n^2} \sum_{i=1}^{n} \text{var}[U_i(\tau_1, \tau_2)] + \frac{1}{n^2} \sum_{i=1}^{n} \sum_{j \neq i}^{n} \text{cov}\left[ U_i(\tau_1, \tau_2), U_j(\tau_1, \tau_2) \right] \\
&= \frac{\tau(T-\tau)}{(T-1)n} (1 + (n-1)\bar{\rho}_n(\tau_1, \tau_2)),
\end{aligned} \tag{22}$$

*where*

$$\bar{\rho}_n(\tau_1, \tau_2) = \frac{1}{n(n-1)} \sum_{i=1}^{n} \sum_{\substack{j=1 \\ j \neq i}}^{n} \rho_{ij}(\tau_1, \tau_2) \tag{23}$$

*is the average cross-sectional correlation of cumulated ranks.*

Cross-sectional dependence affects the asymptotic distribution of the statistic in Equation (20). However, as discussed in (Lehmann 1999, Scttion 2.8), it is frequently true that the asymptotic normality holds provided that the average correlation, $\bar{\rho}_n(\tau_1, \tau_2)$, tends to zero rapidly enough such that

$$\frac{1}{n} \sum_{i \neq j}^{n} \rho_{ij}(\tau_1, \tau_2) = (n-1)\bar{\rho}_n(\tau_1, \tau_2) \to \gamma \text{ as } n \to \infty, \tag{24}$$

where $\gamma$ is some finite constant. Under this condition the limiting distribution of $Z$-statistic in (20) becomes $N(0, 1 + \gamma)$.

Otherwise, from practical point of view, the crucial result of Formula (22) is that the only unknown parameter to be estimated is the average cross-sectional correlation $\bar{\rho}_n(\tau_1, \tau_2)$. Hagnäs and Pynnonen (2014) discuss approaches to account implicitly for this average correlation in cumulated ranks tests when all events share the same calendar day, i.e., the case of complete clustering. These implicit approaches, however, do not work in the case of partial clustering. Therefore, by utilizing the procedure developed in Kolari et al. (2018), this paper proposes a method to estimate explicitly the cross-sectional correlation, $\bar{\rho}_n(\tau_1, \tau_2)$, and thereby solve the cross-sectional correlation problem in the case of the partial clustering.

## 4. Correlation Robust Test for Cumulated Ranks

Following Kolari et al. (2018), let $\tau_{ij}$, $0 \leq \tau_{ij} \leq \tau$ denote the number of calendar days stocks $i$ and $j$ share in common within the event windows. By independence in Assumption 1, the correlation, $\text{cor}\left[ U_{iu}, U_{jv} \right]$, of the standardized ranks $U_{iu}$ and $U_{jv}$ is zero

whenever the underlying calendar days of the relative event days, $u$ and $v$, differ and can be non-zero when the calendar days are the same. Denoting these non-zero correlations (which are also covariances) by $\rho_{ij}$, we get

$$\text{cov}\big[U_i(\tau_1, \tau_2), U_j(\tau_1, \tau_2)\big] = \sum_{u=\tau_1}^{\tau_2} \sum_{v=\tau_1}^{\tau_2} \text{cor}\big[U_{iu}, U_{jv}\big] = \tau_{ij}\rho_{ij}.$$

Combining this with (21), we obtain

$$\rho_{ij}(\tau_1, \tau_2) = \left(\frac{T-1}{T-\tau}\right)\frac{\tau_{ij}}{\tau}\rho_{ij}. \tag{25}$$

We can assume that the overlapping window lengths, $\tau_{ij}$, and the cross-sectional correlations, $\rho_{ij}$, are not dependent on each other so that $\sum_{i \neq j} \tau_{ij}\rho_{ij} = n(n-1)\bar{\tau}\bar{\rho}$, where $\bar{\tau}$ is the average number of overlapping calendar days, and $\bar{\rho}$ is the average cross-sectional correlation of $U_i$ and $U_j$.[3] Consequently, we can rewrite (22) as

$$\text{var}[\bar{U}(\tau_1, \tau_2)] = \frac{\tau(T-\tau)}{(T-1)n}(1 + (n-1)\delta\bar{\rho}), \tag{26}$$

where $\delta = \bar{\tau}(T-1)/(\tau(T-\tau))$ adjusts the average correlation by the fraction of overlapping calendar days within the event window.

It is notable that, even though the average cross-sectional correlation, $\bar{\rho}$, in Equation (26) is based on $n(n-1)/2$ correlations, it can be computed without estimating individual correlations by utilizing the method introduced by Edgerton and Toops (1928). Instead of $n(n-1)/2$ individual correlations, it turns out that one needs to compute only $n+1$ variances, which is a computational problem of order $n$ viz-a-viz of order $n^2$ with averaging the correlations. To illustrate the idea, consider $n$ random variables $x_j$, $j = 1, \ldots, n$ and define the standardized variables $z_j = x_j/\sigma_j$. Next let $\bar{z} = \sum_j z_j/n$ denote the average of the variables. Then because $\text{var}[z_j] = 1$ and $\text{cov}[z_j, z_k] = \text{cor}[z_j, z_k] = \rho_{jk}$, variance of $\bar{z}$ becomes $\sigma_{\bar{z}}^2 = \text{var}[\bar{z}] = (1 + (n-1)\bar{\rho})/n$, we obtain

$$\bar{\rho} = (n\sigma_{\bar{z}}^2 - 1)/(n-1). \tag{27}$$

Hence, to estimate the average cross-sectional correlation, all we need are estimates of $n$ standard deviations of the $x$-variables and the variance of $\bar{z}$. Finally, for large $n$, Equation (27) shows that $\bar{\rho} \approx \sigma_{\bar{z}}^2$.

Because in our case the calendar days of different stocks are only partially overlapping, we estimate the variance of the average utilizing the clustering robust estimation technique (e.g., see Cameron et al. 2011) suggested in Kolari et al. (2018).

Following Kolari et al. (2018), denote the calendar days of the returns in the combined estimation and event window as $t = 1, \ldots, L$, which implies that $L$ becomes the number of clusters equaling the number of separate calendar days on which returns are available in the combined estimation and event windows. Let $n_t$ denote the number of stocks having returns on calendar day $t$ and define

$$U_t = \sum_{k=1}^{n_t} U_{kt}. \tag{28}$$

Then

$$U_t^2 = \sum_{k=1}^{n_t} U_{kt}^2 + \sum_{i \neq j}^{n_t} U_{it}U_{jt}, \tag{29}$$

so that

$$\sum_{i \neq j}^{n_t} U_{it}U_{jt} = U_t^2 - \sum_{k=1}^{n_t} U_{kt}^2. \tag{30}$$

Summing these up over the calendar days in the combined estimation and event window, we have

$$\sum_{t=1}^{L}\sum_{i\neq j}^{n_t} U_{it}U_{jt} = \sum_{t=1}^{L} U_t^2 - \sum_{t=1}^{L}\sum_{k=1}^{n_t} U_{kt}^2. \tag{31}$$

Taking the average, we get an estimator for the average correlation

$$\hat{\rho} = \frac{1}{M}\sum_{t=1}^{L}\sum_{i\neq j}^{n_t} U_{it}U_{jt}, \tag{32}$$

where

$$M = \sum_{t=1}^{L} n_t(n_t - 1) \tag{33}$$

is the number of the cross-product terms. It is notable that days for which there is available only one return drop automatically out (if $n_t = 1$ for all $t$, then $\hat{\rho} = 0$). The potentially tedious computation over all cross-products can be materially simplified by utilizing the right-hand- side of Equation (31). By Result 2 the variances of standardized ranks are all equal to one and means equal zero. Therefore, arranging the terms of the rightmost sum of Equation (31) to correspond to variance representations, the (double) sum becomes equal to $\sum_{t=1}^{L} n_t$, i.e., the total number of observations.[4] Thus, the only component we need to compute is the first sum of squares on the right-hand-side of (31). Therefore, similar to the illustration of computing the average correlation above, the computational effort of computing the average correlation is again of order $n$ (rather than $n^2$). Finally, we get:

**Result 5.** *A computationally efficient form of the average correlation in* (32) *is*

$$\hat{\rho} = \frac{N}{M}(s_U^2 - 1), \tag{34}$$

*where $N = \sum_{t=1}^{L} n_t$ is the total number of returns, M is given by* (33), *and*

$$s_U^2 = \frac{1}{N}\sum_{t=1}^{L} U_t^2 \tag{35}$$

*with $U_t$ given in Equation* (28). *Variance, $s_U^2$, is a clustering robust variance estimator of standardized ranks in the presence of intra-cluster correlation (cf. e.g.,* Cameron et al. 2011).

As noted earlier, $\hat{\rho} = 0$ if all $n_t = 1$.

Given the estimator of the average cross-sectional correlation, $\bar{\rho}$, we can define an appropriate cross-sectional correlation robust test for the null hypothesis of zero cumulative abnormal returns

$$H_0 : \mu(\tau_1, \tau_2) = \mathbb{E}[\text{CAR}(\tau_1, \tau_2)] = 0. \tag{36}$$

The test can be defined in terms of the cumulated ranks using the $z$-ratio

$$z_\tau = \frac{\bar{U}(\tau_1, \tau_2)}{\sigma_\tau\sqrt{1 + (n-1)\delta\hat{\bar{\rho}}}}, \tag{37}$$

where $\sigma_\tau$ is the square root of Equation (19), i.e., the variance

$$\sigma_\tau^2 = \frac{\tau(T - \tau)}{(T - 1)n}$$

of $\bar{U}(\tau_1, \tau_2)$ for completely non-overlapping event windows in calendar time [i.e., when $\bar{\rho} = 0$ in Equation (26)], and $\tau = \tau_2 - \tau_1 + 1$ is the length of the window of cumulated abnormal returns.

In event studies, the combined length, $T$, of the estimation and event period remains fixed, while the number of event firms, $n$, defines the sample size, thereby being the dimension increased when dealing with the asymptotic distribution of associated test statistics.

Given that the condition in Equation (24) holds for $\hat{\rho}$, the null distribution of $z_\tau$ is asymptotically normal with zero mean and unit variance.

Kolari and Pynnonen (2011) propose replacing the cumulative ranks in Definition 2 by a single rank number which is based on standardized cumulative abnormal returns (SCARs)

$$\text{SCAR}_i(\tau_1, \tau_2) = \frac{\text{CAR}_i(\tau_1, \tau_2)}{S_{\text{CAR}_i(\tau_1,\tau_2)}} \tag{38}$$

in which $S_{\text{CAR}_i(\tau_1,\tau_2)}$ is the standard deviation of $\text{CAR}_i(\tau_1, \tau_2)$ (for details, see Kolari and Pynnonen 2011). Their approach again accounts implicitly for cross-sectional correlation due to completely overlapping event days. Here, we extend the approach to cover the partial overlapping case. Rather than using the scaled ranks defined in Kolari and Pynnonen (2011), we use the standardized ranks of Definition 1. Subsequently, denoting the standardized rank of $\text{SCAR}_i(\tau_1, \tau_2)$ by $U_{i0}$, we can base the rank test for testing the null hypothesis of zero cumulative abnormal returns in Equation (36) on the average ranks

$$\bar{U}_0 = \frac{1}{n} \sum_{i=1}^{n} U_{i0}. \tag{39}$$

If the event periods are completely non-overlapping, $U_{i0}$s are independent with zero mean and unit variance (see Definition 1), in which case the null distribution of $\bar{U}_0$ has zero mean and variance $1/n$. However, if the event days are partially overlapping, the components of $\bar{U}_0$ absorb the cross-sectional correlation over the CAR-window. The correlation that inflates the variance is inherited from the cross-sectional correlations of $\text{SCAR}_i$s. Kolari et al. (2018) show that the variance inflation factor is of the form $(1 + (n-1)\nu\bar{\rho})$ as in Equation (26) with the exception that $\delta$ is replaced by $\nu = \bar{\tau}/\tau$, the ratio of the average number of overlapping calendar days within the CAR-window to the window length. With this correction the variance of $\bar{U}_0$ becomes $\text{var}[\bar{U}_0] = (1 + (n-1)\nu\bar{\rho})/n$. We can estimate the average cross-sectional correlation, $\bar{\rho}$, as in Equation (32) utilizing only the estimation period in computing $s_U^2$. For this approach, the standardized ranks in Definition 1 are redefined for the estimation period abnormal returns. Alternatively one can estimate the cross-sectional correlation exactly as in Result 5. Both approaches will produce essentially the same result in most cases. With the estimated average correlation, we get a cross-sectional correlation robust generalized rank test statistic

$$z_{\tau,\text{grank}} = \frac{\sqrt{n}\,\bar{U}_0}{\sqrt{1 + (n-1)\nu\hat{\rho}}}, \tag{40}$$

where $\nu = \bar{\tau}/\tau$. Again, given that the condition in Equation (24) holds for $\hat{\rho}$, the null distribution of $z_{\tau,\text{grank}}$ is asymptotically normal with zero mean and unit variance.

## 5. Simulation Results

We generate artificial returns utilizing the Fama and French (2015) five-factor model (FF5),

$$(r_{it} - r_f)_t = \alpha_i + \beta_{i,\text{mkt}}(r_m - r_f)_t + \beta_{i,\text{smb}}\text{SMB}_t + \beta_{i,\text{hml}}\text{HML}_t + \beta_{i,\text{rmw}}\text{RMW}_t + \beta_{i,\text{cmw}}\text{CMW}_t + \epsilon_{it}, \tag{41}$$

where $r_m - r_f$ is the market excess return over the risk-free rate $r_f$, SMB, HML, RMW, and CMW are common market factors proposed by Fama and French. We utilize daily data from 2 January 1990 through 30 October 2020 (7770 daily returns) to generate 20,000 initial daily return series for this sample period. The regression coefficients for each stock are generated from multivariate normal distribution with mean vector $(0, 1, 0.5, 0.5, 0.5, 0.5)$

and covariance matrix $\sigma_i^2(X'X)^{-1}$, in which $\sigma_i^2$ is the variance of the error term $\epsilon$. The stock specific $\sigma_i^2$ values are generated by drawing $\sigma_i$s, the standard deviations, independently from a uniform distribution $U(1,3)$. This corresponds to a range of annual volatilities roughly from 10 percent to 48 percent. The $(X'X)$ matrix is the cross-product matrix of the Fama-French 5-factor regression model.[5] The (7770) error terms $\epsilon_{it}$ for stock $i$ is generated independently from the normal distribution $N(0, \sigma_i^2)$.

In the simulations we define the abnormal returns with respect to the market model as

$$AR_{it} = (r_i - r_f)_t - (\hat{\alpha}_i + \hat{\beta}_i(r_m - r_f)_t), \tag{42}$$

where $\hat{\alpha}_i$ and $\hat{\beta}_i$ are ordinary least squares (OLS) estimates. Therefore, missing common factors introduce cross-sectional correlation between the abnormal returns. The event period is $\pm 10$ trading days around the event day $t = 0$, and the estimation period consists of 250 days prior the event periods, i.e., relative days $-260, \ldots, -11$.

In forthcoming experiments we focus on the effect of cross-sectional correlation on the size of the test. Other issues, such as event induced volatility are well documented for example by Kolari and Pynnonen (2011); Kolari and Pynnönen (2010). Utilizing the base design initiated by Brown and Warner (1985), we generate 1000 samples of randomly selected 50 stocks (the returns of which are generated by the FF5 model in Equation (41)) with four over-lapping event days scenarios. In the first case of non-overlapping event days, the returns are cross-sectionally independent. In the second case of completely overlapping events, all firms share the same event day (calendar time), and in the third and fourth scenarios the event days are randomly scattered across 5 and 10 concecutive calendar days, i.e., one and two weeks of trading days, respectively.

We report two-tailed rejection rates for the null hypothesis of no event-effect across different event windows of $\pm 1$, $\pm 2$, $\pm 5$, and $\pm 10$ around the event day, i.e., window lengths $\tau = 1, 3, 5, 10$, and 21 days. In addition to statistic $z_\tau$ in Equation (37) we report results for the more traditional rank based test proposed by (Campbell and Wasley 1993, p. 85):

$$z_{cw} = \frac{\sum_{t=\tau_1}^{\tau_2} \bar{k}_t}{\sqrt{\tau} s_{\bar{k}}}, \tag{43}$$

where

$$\bar{k}_t = \frac{1}{n} \sum_{i=1} (K_{it} - \mathbb{E}[K_{it}]) \tag{44}$$

with $\mathbb{E}[K_{it}] = (T+1)/2$ and

$$s_{\bar{k}}^2 = \frac{1}{T} \sum_{t=T_0+1}^{T_2} \bar{k}_t^2. \tag{45}$$

Furthermore, we report results for traditional parametric (cross-sectional correlation non-robust) *t*-statistics popular in event studies (e.g., see (Campbell et al. 1997, chp. 4)),

$$t_\tau = \frac{\overline{CAR}(\tau_1, \tau_2)}{\text{s.e.}(CAR)}, \tag{46}$$

where $\overline{CAR}(\tau_1, \tau_2)$ is the sample average of $CAR_i(\tau_1, \tau_2)$ defined in (2), and s.e.(CAR) is the related standard error. Under independence, the null distribution of $t_\tau$ is asymptotically standard normal.

Table 1 summarizes the test statistics and their major features.

Table 2 reports the simulation results of the two-tailed rejection rates of the null hypothesis of no abnormal return at the 5% nominal rejection rate. The results are clear-cut. Panel A of the table reports the non-overlapping case with zero cross-sectional correlation. As expected, all statistics reject close to the nominal rate. Panel B reports results of complete overlapping. That is, all events share the same calendar day; hence, returns are prone to cross-sectional correlation. The new $z_\tau$, $z_{\tau,\text{grank}}$, and the more traditional cumulative

ranks statistic, $z_{cw}$, that account for cross-sectional correlation, reject reasonably close to the nominal rate up to event windows $\pm 5$ and exhibit some over-rejection on the longest event window $\pm 10$, i.e., 21 days. Not surprisingly, the parametric, non-cross-correlation robust statistic, $t_\tau$, incrementally over-rejects as event windows increase in length. Panel C reports partial overlapping with events clustered randomly within 5 trading days (about a week). For event day testing also the a priori non-robust statistics perform well by rejecting at the nominal rate. However, they start to incrementally over-reject as the event window grows longer. The a priori partial overlapping robust statistics, $z_\tau$ and $z_{\tau,\text{grtank}}$, reject close to the nominal rate up to the event window lengths of 5 days and over-reject to some extent for the longest event windows of 11 and 21 days, albeit far less than the non-robust statistics of $z_{cw}$ and $t_\tau$. The results are pretty much similar with the decreased overlapping in Panel D. Thus, we conclude that accounting for cross-sectional correlation is crucial to avoid biased inferences in statistical testing, not only due to complete overlapping of event windows, but also for partially overlapping cases. Regarding the latter, this paper has introduced two new test statistics that account for these cases.

**Table 1.** Test statistics and their key features.

| | | | | Robustness Due to | |
| | | | | Correlation Caused by | |
| Statistic | Type | Event Volatility | Complete Ovrlp | Partial Ovrlp |
| --- | --- | --- | --- | --- |
| $z_\tau = \frac{\bar{U}(\tau_1,\tau_2)}{\sigma_\tau \sqrt{1+(n-1)\delta\hat{\rho}}}$, Equation (37) | non-parametric | yes | yes | yes |
| $z_{\tau,\text{grank}} = \frac{\sqrt{n}\,\bar{U}_0}{\sqrt{1+(n-1)\nu\hat{\rho}}}$, Equation (40) | non-parametric | yes | yes | yes |
| $z_{cw} = \frac{\sum_{t=\tau_1}^{\tau_2} \bar{k}_t}{\tau s_{\bar{k}}}$, Equation (43) | non-parametric | no | yes | no |
| $t_\tau = \frac{\overline{\text{CAR}}(\tau_1,\tau_2)}{\text{s.e}(\overline{\text{CAR}})}$, Equation (46) | parametric | yes | no | no |

**Table 2.** Rejection rates of the null hypothesis of no event effect at the nominal 5% level when the events are no-overlapping, partially overlapping, and completely overlapping.

| | CAR Window Length | | | | |
| --- | --- | --- | --- | --- | --- |
| | 1<br>Event Day | 3<br>$(-1,+1)$ | 5<br>$(-2,+2)$ | 11<br>$(-5,+5)$ | 21<br>$(-10,+10)$ |
| Panel A: Non-clustered events | | | | | |
| $z_\tau$ | 0.048 | 0.054 | 0.050 | 0.052 | 0.064 |
| $z_{\tau,\text{grank}}$ | 0.048 | 0.052 | 0.053 | 0.058 | 0.053 |
| $z_{cw}$ | 0.052 | 0.050 | 0.051 | 0.052 | 0.063 |
| $t_\tau$ | 0.045 | 0.035 | 0.049 | 0.052 | 0.048 |
| Panel B: Events clustered on the same trading day | | | | | |
| $z_\tau$ | 0.059 | 0.051 | 0.059 | 0.064 | 0.072 |
| $z_{\tau,\text{grank}}$ | 0.064 | 0.055 | 0.065 | 0.067 | 0.082 |
| $z_{cw}$ | 0.059 | 0.052 | 0.061 | 0.064 | 0.075 |
| $t_\tau$ | 0.087 | 0.091 | 0.096 | 0.085 | 0.110 |
| Panel C: Events clustered on 5 consecutive trading days | | | | | |
| $z_\tau$ | 0.056 | 0.055 | 0.059 | 0.086 | 0.076 |
| $z_{\tau,\text{grank}}$ | 0.056 | 0.057 | 0.066 | 0.083 | 0.075 |
| $z_{cw}$ | 0.050 | 0.075 | 0.093 | 0.127 | 0.129 |
| $t_\tau$ | 0.045 | 0.063 | 0.077 | 0.112 | 0.102 |
| Panel D: Events clustered on 10 consecutive trading days | | | | | |
| $z_\tau$ | 0.056 | 0.055 | 0.059 | 0.086 | 0.076 |
| $z_{\tau,\text{grank}}$ | 0.064 | 0.046 | 0.064 | 0.065 | 0.082 |
| $z_{cw}$ | 0.059 | 0.062 | 0.091 | 0.116 | 0.133 |
| $t_\tau$ | 0.065 | 0.057 | 0.056 | 0.089 | 0.105 |

## 6. Summary and Conclusions

This paper proposed two variants of a new non-parametric rank based test statistic for testing cumulative abnormal returns in short-run event studies. The statistics are robust to event-induced volatility and cross-sectional correlation due to complete or partially overlapping event windows. This latter source of cross-sectional correlation is not taken into account by the existing non-parametric test statistics. Simulation results indicate that, unlike typically utilized test statistics, the proposed statistics reject the null hypothesis of no event effect close to the nominal significant level in the partially overlapping case. We conclude that accounting cross-sectional correlation is crucial to avoid biased inferences, not only due to complete overlapping of event windows but also for partial overlapping cases. The non-parametric test statistics proposed in this paper serve this purpose. A major limitation of utilizing non-parametric tests in financial economics is that they seem to play mainly side roles. For example, (Campbell et al. 1997, Sction 4.7) note that non-parametric tests are typically used in conjunction with parametric tests to check robustness of conclusions based on parametric tests. Even so, it should be noted that robustness checks are incrementally demanded in modern empirical financial research. Non-parametric methods can be the tools of choice in completing the task.

**Funding:** The research has not received external funding.

**Institutional Review Board Statement:** Not applicable.

**Informed Consent Statement:** Not applicable.

**Data Availability Statement:** Data available upon request from the author.

**Acknowledgments:** The author wants to thank James Kolari, Henk Snoo, Rudi Wietsma, and referees for many useful comments that improved the paper. All errors are the responsibility of the author.

**Conflicts of Interest:** The author declares no conflict of interest.

## Notes

[1] For discussion of true abnormal returns, see Harrington and Shrider (2007).

[2] We define conventional event studies as those focusing only on mean stock price effects. Other types of event studies include (for example) the examination of return variance effects (Beaver (1968); Patell (1976)), trading volume (Beaver (1968); Cambell and Wasley (1996)), accounting performance (Barber and Lyon (1997)), and earnings management procedures (Dechow et al. (1995); Kothari et al. (2005)).

[3] The equation follows by setting $\sum(x - \bar{x})(y - \bar{y}) = \sum xy - n\bar{x}\bar{y}$ to zero, so that $\sum xy = n\bar{x}\bar{y}$.

[4] That is,

$$\sum_{t=1}^{L}\sum_{k=1}^{n_t} U_{kt}^2 = \sum_{t=t_1}^{L_1} U_{1t}^2 + \sum_{t=t_2}^{L_2} U_{2t}^2 + \cdots + \sum_{t=t_n}^{L_n} U_{nt}^2 = \sum_{i=1}^{n}\sum_{t=t_i}^{L_i} U_{it}^2,$$

where $t_i, t_i + 1, \ldots, L_i$ indicate observations on stock $i$ with $T_i = L_i - t_i + 1$, the number of observations. By Result 2 $\text{var}[U_{it}] = 1$, so that $\sum_{t=t_i}^{L_i} U_{it}^2 = T_i$. Hence, $\sum_{t=1}^{L}\sum_{k=1}^{n_t} U_{kt}^2 = \sum_{i=1}^{n} T_i = N = \sum_{t=1}^{L} n_t$.

[5] Factor returns have been downloaded from the French data library. http://mba.tuck.dartmouth.edu/pages/faculty/ken.french/data_library.html, accessed on 15 November 2021.

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
