# Peer review of "Non-Parametric Statistic for Testing Cumulative Abnormal Stock Returns"

_jrfm, doi:10.3390/jrfm15040149_

Round 1

Reviewer 1 Report

In this paper, a new non-parametric rank based test statistic for testing cumulative abnormal returns is proposed. The modification of existing approaches is made to improve the robustness to cross-sectional correlation of returns due to calendar time partially overlapping event windows. The methodology is tested on simulated returns utilizing Fama and French five-factor model.

My concerns are as follows:

  • The review of related literature is comparatively short. Whether this indicates that little research in the field has been carried so far? Or maybe the problem of event identification is not so relevant issue?
  • The day t=0 (Page 2) is indicated as the event day. Why is it defined by setting it as initial time moment?
  • Below the Eq.6 it should be SARit instead of SARits (Page 4).
  • Below The Eq.14 in the variance-of-the-sum formula it should be τ1 instead of t1 in the summation operator (Page 5).
  • In the formula of variance-of-the-sum (Page 5), shouldn’t it be a covariance multiplied by two, which then is referred in deriving Result 4?
  • Some derivations of provided results (for example, Eq.32-35 equations, Eq.40) is not so straightforward and includes some interim rearrangements, therefore it is recommended to include the appendix to show the derivation (proof).
  • To my mind, the paper would benefit if the figures would be included in the paper demonstrating the partial clustering, complete clustering, abnormal returns, and other time series features considered in the paper in order to demonstrate the concepts used in the paper. Then, more importantly, the visual explanation of identified events using simulated data is also highly recommended.
  • The section of conclusions is like a summary of what has been presented in the paper. The author could discuss the practical recommendations of their approach, some issues (if any) that may arise during the application, some limitations, and also future directions.
  • The paper should be formatted according to the journal’s requirements if accepted.

Author Response

Many thanks to the reviewer for the numerous useful comments. Please see the attachment.

Reviewer 2 Report

Amazing well-written fascinating.

Author Response

Many thanks for the careful comments and suggestions.

(1) As proposed by the reviewer we have revised the title of the paper to: "Proposed Non-parametric Statistic for Testing Cumulative Abnormal Stock Returns"

(2) My native English (American) speaking colleague has kindly helped me with the languages of the manuscript.

Round 2

Reviewer 1 Report

The revised manuscript has been considerably improved, and most of concerns have been addressed. My current comments are as follows:

  • I suggest to reconsider the title of manuscript, excluding the term “proposed” or changing it;
  • Footnote is preferred for a short comment, while the proof, which requires one third of page, could be moved to appendix;
  • In the title and in the introduction the importance of non-parametric testing of abnormal stock returns is presented. However, after the derivation of non-parametric statistic and demonstrating its calculation by providing a simulated example, no discussion can be found how it could be helpful for risk management purposes or asset pricing, reflecting the journal’s scope. Furthermore, the question is whether it could be applied for stock returns only and why?
  • In the first report the last comment about manuscript formatting according to journal’s requirements was ignored and no response received.

Author Response

We want to thank the reviewer for the great comments. Please find below our responses, and please find the related edits in the uploaded manuscript.

Reviewer: I suggest to reconsider the title of manuscript, excluding the term “proposed” or changing it;

Response: "proposed dropped"

Reviewer: Footnote is preferred for a short comment, while the proof, which requires one third of page, could be moved to appendix;

Response: We fully agree that footnotes if used should be short notes. Our motivation to avoid technical appendices is to keep the presentation compact. Of course if it is absolute necessity to add a more verbose 'proof' of the result, we are glad to do it (e.g. web appendix?).  Observed that the proposed version was indeed far too verbose as we have in the text already referred to the main background. Now we have reduced the footnote  (footnote 5) to contain only the essential point with the consequence, hence essentially 3 lines. Believe this should at one glimpse reveal the idea communicated in the main text (so no need for the interested reader to go appendices).

Reviewer: In the title and in the introduction the importance of non-parametric testing of abnormal stock returns is presented. However, after the derivation of non-parametric statistic and demonstrating its calculation by providing a simulated example, no discussion can be found how it could be helpful for risk management purposes or asset pricing, reflecting the journal’s scope. Furthermore, the question is whether it could be applied for stock returns only and why?

Response: We have elaborated the discussion of the importance of our test statistic in the first paragraph of the introduction (marked red). Referring in particular to Fama (1991) we have motivated the importance of event studies and reliable empirical tools in related statistical testing for which our proposed tests.should be useful. We refer also here and also elsewhere 'asset returns'. That is, bonds, currencies, options, etc. are covered, even though the major research so far has been on stock markets (also bond markets are covered to some extend).

Reviewer: In the first report the last comment about manuscript formatting according to journal’s requirements was ignored and no response received.

Response: This will be done if the manuscript will be accepted. In the review process, we have tried to facilitate ease of reading with larger fonts, etc.

Round 3

Reviewer 1 Report

The revised paper is considerably improved, and most of concerns have been addressed.

The formatting of the paper according to journal's requirements is required.